# Enhancing Adversarial Defense via Brain Activity Integration Without Adversarial Examples

**DOI:** 10.3390/s25092736

**Published:** 2025-04-25

**Authors:** Tasuku Nakajima, Keisuke Maeda, Ren Togo, Takahiro Ogawa, Miki Haseyama

**Affiliations:** 1Graduate School of Information Science and Technology, Hokkaido University, N-14, W-9, Kita-ku, Sapporo 060-0814, Japan; nakajima@lmd.ist.hokudai.ac.jp; 2Faculty of Information Science and Technology, Hokkaido University, N-14, W-9, Kita-ku, Sapporo 060-0814, Japan; maeda@lmd.ist.hokudai.ac.jp (K.M.); togo@lmd.ist.hokudai.ac.jp (R.T.); ogawa@lmd.ist.hokudai.ac.jp (T.O.)

**Keywords:** adversarial defense, brain activity, CLIP model, data augmentation

## Abstract

Adversarial attacks on large-scale vision–language foundation models, such as the contrastive language–image pretraining (CLIP) model, can significantly degrade performance across various tasks by generating adversarial examples that are indistinguishable from the original images to human perception. Although adversarial training methods, which train models with adversarial examples, have been proposed to defend against such attacks, they typically require prior knowledge of the attack. These methods also lead to a trade-off between robustness to adversarial examples and accuracy for clean images. To address these challenges, we propose an adversarial defense method based on human brain activity data by hypothesizing that such adversarial examples are not misrecognized by humans. The proposed method employs an encoder that integrates the features of brain activity and augmented images from the original images. Then, by maximizing the similarity between features predicted by the encoder and the original visual features, we obtain features with the visual invariance of the human brain and the diversity of data augmentation. Consequently, we construct a model that is robust against adversarial attacks and maintains accuracy for clean images. Unlike existing methods, the proposed method is not trained on any specific adversarial attack information; thus, it is robust against unknown attacks. Extensive experiments demonstrate that the proposed method significantly enhances robustness to adversarial attacks on the CLIP model without degrading accuracy for clean images. The primary contribution of this study is that the performance trade-off can be overcome using brain activity data.

## 1. Introduction

Large-scale vision–language models (LVLMs) such as LLaVA [1] possess advanced multimodal processing abilities for both visual and language information, achieving exceptional performance in various downstream tasks such as image generation and captioning. When constructing an LVLM, the CLIP model [2], which is trained on a large dataset of image–text pairs, is typically used as the foundational model. The CLIP model acquires highly expressive capabilities across modalities in a shared embedding space constructed by contrastive learning [3] between two modalities.

The CLIP model, with its highly expressive capabilities, is effectively used for various tasks; however, it is vulnerable to unexpected attacks [4,5]. For example, an adversarial attack method can generate subtly modified examples that are indistinguishable from the original clean images by human perception [6]. However, if these examples are input into the CLIP model, it causes the model to produce incorrect outputs, which severely affects various downstream tasks such as zero-shot classification, image generation, and caption generation [7,8,9]. These examples are referred to as adversarial examples, and numerous researchers have actively investigated defense methods to combat them. In particular, adversarial training [10], which uses adversarial examples as part of the training dataset, has exhibited high defense performance. Despite these efforts, if the CLIP model is trained to be robust to adversarial examples, it overfits these special input patterns, leading to the degradation of accuracy for nonattacked images (clean images). Therefore, there is a trade-off between the robustness of a model to adversarial attacks and its accuracy for clean images, and a new defense method that can achieve both goals is required.

Recent studies have demonstrated that self-supervised learning (SSL) improves robustness to adversarial attacks [11]. The reason is that by learning the consistency of data representation under several different data augmentations, a label-independent model that obtains diverse features can be developed. Inspired by this finding, an increasing number of researchers have begun to incorporate SSL into adversarial defense methods [12,13]. However, most such models combine it with adversarial training to improve robustness because SSL alone cannot fully capture the characteristics of perturbations in adversarial examples. Therefore, it is necessary to construct a new method that incorporates the data augmentation ability of SSL.

In recent years, research into applying human brain activity to machine learning models for image processing has attracted increasing attention [14,15,16]. When recognizing images, humans use functions such as memory and decision-making [17]. This process allows humans to accurately recognize objects and describe situations without being misled by slight changes in appearance features, such as size, position, and color, or by adversarial attacks. This cognitive system functions comprehensively through neural activities spanning multiple areas of the brain, such as the frontal and temporal lobes, and closely coordinates with brain activity. In other words, humans recognize information based on the ability of the brain to encode visual stimuli through complex neural networks [18]. Brain activity with complex processing capabilities is a factor that prevents humans from being affected by adversarial attacks. A previous study [19] on adversarial defense using brain activity demonstrated an improvement in robustness against adversarial attacks using simple image classification models, such as handwritten character recognition. Therefore, it is expected that the above problems can be solved by incorporating human brain activities into the adversarial defense method.

In this study, we propose an adversarial defense method based on brain activity data to mitigate adversarial attacks targeting the CLIP model. Inspired by the fact that humans are unaffected by adversarial attacks, we introduce an encoder that outputs visual features that are robust against changes in images using brain activity data and augmented images. Assuming that the human brain exhibits similar activity for an image even when slight transformations are applied, we integrate the features of brain activity data with those of slightly transformed images obtained through data augmentation. Next, we develop a model that embeds features in the CLIP space by maximizing the similarity between features predicted by the encoder and the original features. As shown in Figure 1, the proposed method does not use information about adversarial attacks during training; thus, it is highly effective in terms of addressing the degradation of accuracy for clean images. To the best of our knowledge, the proposed method is the first adversarial defense method for the CLIP model that uses brain activity data and does not depend on specific attack methods. Extensive experiments demonstrate the effectiveness of brain activity data against adversarial attacks targeting the CLIP model and the superiority of the proposed method. The primary contribution of this study is the demonstration of the effectiveness of using brain activity data to address the challenges associated with adversarial defense.

## 2. Related Work

In this section, we present related studies on CLIP and adversarial attacks to clarify the novelty and primary contribution of this study.

### 2.1. Contrastive Language–Image Pretraining

In conventional computer vision, supervised learning using human-defined labels has been the mainstream approach [20,21]. However, this approach has some challenges, such as the high cost of collecting large-scale labeled datasets and low generalizability to new classes. To address these challenges, this study proposes a method that leverages natural language to train vision models without relying on labeled datasets. Specifically, a dataset of 400 million (400 M) image–text pairs collected from the Internet is used to train an image encoder and a text encoder in a shared embedding space. Through contrastive learning, the cosine similarity between images and text is maximized, enabling the acquisition of a unified and generalizable embedding representation that integrates visual and language information.

Zero-shot classification refers to a task where a model classifies objects it has never seen during training. CLIP, which uses contrastive learning with a large amount of natural language data, enables general-purpose classification without relying on predefined labels. This technology allows CLIP to compare and understand images and text, facilitating applications in image retrieval, image generation, object detection, and various other fields that integrate visual and language information [22,23,24].

### 2.2. Adversarial Attacks

An adversarial attack is a type of attack that can be applied to machine learning models. This attack adds perturbations to the input sample to increase the error with the correct label and creates an example that causes the model to misclassify [25,26]. The types of attacks can be classified as white-box attacks [27,28,29] and black-box attacks [30,31,32].

In a white-box attack, the attacker has complete knowledge of the parameters, architecture, and gradient information of the target model and can perform a precisely designed attack. Thus, the success rate of the attack is high. However, in real-world scenarios, obtaining detailed information about a target model is difficult; thus, the practicality of white-box attacks is low. In contrast, a black-box attack is performed with only the target model’s output known. A black-box attack relies on gradient estimation and perturbation search; thus, its success rate is lower than that of a white-box attack. However, because it can be applied without knowing the details of the target model, its practicality is high.

## 3. Proposed Method

In this section, we explain the proposed adversarial defense method using brain activity. An overview of the proposed method is presented in Figure 2. The proposed method involves three steps. First, in Section 3.1, we describe how to create an augmented image set from the original image. Next, in Section 3.2, we explain how to extract and integrate features from both the augmented image set and brain activity data. Finally, in Section 3.3, we describe the training of the proposed method.

### 3.1. Creation of Augmented Image Set

To represent the characteristics of perturbations in adversarial examples without using the examples, we obtained diverse features from a single image. Specifically, we applied several data augmentation techniques to the original image $X^{o}$. The types of data augmentation techniques were based on the transformations employed in SimCLR [33], which is a widely used SSL model. We used four types of augmentations. Specifically, we used “augmentation includes cropping, resizing, and flipping” and “augmentation includes rotation and horizontal translation” as spatial/geometric transformations of the data. In addition, we used “augmentation to grayscale” and “augmentation to put in Gaussian blur” as the appearance transformation of the data. We obtained the augmented image set $X^{a}$ consisting of the four images transformed using these data augmentations and the original image $X^{o}$.

The above process yielded diverse features to represent the characteristics of perturbations in adversarial examples from only a single original image. In addition, it is expected to obtain a model that does not degrade accuracy for clean images, even for datasets containing brain activity for which a small amount of data are obtained.

### 3.2. Feature Extraction and Integration

To integrate the visual information from the augmented image set $X^{a}$ with the brain activity data, we first extracted features from each data. We used functional magnetic resonance imaging (fMRI) data *Y* of brain activity captured by fMRI [34] when the subjects viewed the original images $X^{o}$. Specifically, we extracted visual features $F^{a}=\left[f_{1}^{a},f_{2}^{a},\cdots ,f_{N}^{a}\right]\in R^{D\times N}$ from the augmented image set $X^{a}$ and brain features $f^{b}\in R^{D}$ from the fMRI data *Y*. Here, *D* represents the dimension in the CLIP space, and *N* represents the number of images in the augmented image set. In the proposed method, *N* is five, as described in Section 3.1. The visual features for the original and augmented images were extracted using the trained CLIP image encoder C(·) using the Vision Transformer (ViT-L/14) [2], and the brain features were extracted using an fMRI decoder M(·). The fMRI decoder, specifically the MindBridge model [35], is designed to predict visual information related to the images viewed from the fMRI data, and it can embed these features in the CLIP space. The extracted features are obtained as follows:(1)$$F^{a}=C\left(X^{a}\right),\;f^{b}=M\left(Y\right).$$
For loss calculation, we integrated the extracted features. We concatenated the feature vectors of the augmented image set and the brain activity data to create the integrated features $F^{i}=\left[f_{1}^{i},f_{2}^{i},\cdots ,f_{N}^{i}\right]\in R^{2D\times N}$ as follows:(2)$$f_{n}^{i}={\left[\begin{array}{c} f^{b^{⊤}},{f_{n}^{a}}^{⊤} \end{array}\right]}^{⊤}\;\left(n=1,2,\cdots ,N\right).$$

The above process yielded integrated features for learning diverse characteristics from the data augmentation and invariant characteristics of the human brain. These features make the model robust against adversarial attacks without adversarial examples.

### 3.3. Training Process

We construct a three-layer multilayer perceptron (MLP) $g_{\theta }\left(\cdot \right)$ to predict the original visual features $f^{o}=C\left(X^{o}\right)\in R^{D}$ in the CLIP space using the integrated features $F^{i}$ described in Section 3.2. The parameters θ of the MLP were optimized by minimizing the loss L to maximize the cosine similarity between the predicted features ${\hat{F}}^{o}=\left[{\hat{f}}_{1}^{o},{\hat{f}}_{2}^{o},\cdots ,{\hat{f}}_{N}^{o}\right]\in R^{D\times N}$ and the original visual features $f^{o}$. The loss function is calculated as follows:(3)$$L=\frac{1}{N}\sum_{n=1}^{N}\left(1-\frac{f^{o^{⊤}}{\hat{f}}_{n}^{o}}{|f^{o}\left|\right|{\hat{f}}_{n}^{o}|}\right).$$
In the proposed method, we minimize the total loss by summing L for all training original images.

Consequently, we constructed a CLIP feature embedding model that uses brain activity. The model embeds robust features against adversarial attacks. In addition, because the proposed method does not require adversarial examples, it is independent of attack type and does not degrade accuracy for clean images.

During testing, the integrated features, which combine the features extracted by CLIP from the test images and the corresponding brain visual features, were input to the trained MLP.

## 4. Experiments

### 4.1. Experimental Settings

#### 4.1.1. Dataset

We used the natural scene dataset (NSD) [36] as the brain activity dataset. Figure 3 shows examples of the dataset. The NSD contains a pair of the original images collected from the MSCOCO dataset [37] and their corresponding fMRI data. We used 9841 samples from a single subject from the NSD, of which 8859 samples were training data and 982 samples were test data. This setting followed the procedure in a previous study [35].

#### 4.1.2. Attack Settings

In this study, four types of gradient-based adversarial attacks were used to evaluate robustness against adversarial attacks, mitigate accuracy degradation for clean images, and reduce the dependence on the attack type. Conventional adversarial defense methods [38,39] typically evaluate performance using two or three types of adversarial attacks. Therefore, our evaluation using four types of attacks is reasonable. In particular, in addition to the commonly used FGSM-derived adversarial attacks PGD [27], APGD [28], and CW [29], we used the state-of-the-art attack method [40], which reduces the success rate of attacks but is generally attackable for any model. For each attack, the goal was to minimize the cosine similarity between the original image and the annotated caption in the CLIP space. The perturbation was applied with $l_{\infty }$ norms of ϵ=1/255 and ϵ=2/255 which are commonly used to evaluate adversarial robustness [10,12]. Here, ϵ represents the intensity of the perturbation.

#### 4.1.3. Evaluation Details

To evaluate the effectiveness of the proposed method (PM), both quantitative and qualitative evaluations were conducted.

**Comparison Methods.** To the best of our knowledge, no previous adversarial defense for the CLIP model requires neither adversarial examples nor degrades accuracy for clean images while being independent of attack types. Therefore, PM cannot be directly compared with other methods. In particular, several studies [41,42] that have considered new tasks do not adopt comparative methods; rather, they have conducted extensive experiments. Therefore, we also verified as many different perspectives as possible in the four comparison experiments. We used the following models:


**CLIP**
This method involves the CLIP model with trained VIT-L/14. Adversarial examples are designed to deceive the model; thus, the output of this method indicates the success rate of the adversarial attack.
**PM w/o fMRI**
This method involves the MLP model trained on only the augmented image set without fMRI data. By comparing this method with PM, we evaluated the effectiveness of brain activity.
**PM w/o fMRI with noise**
This method involves the MLP model trained on random noise data rather than fMRI data. This method represents the chance level, and by comparing brain activity with the chance level, we verified the significance of brain activity.
**RS:Randomized Smoothing [43]**
This method is a type of certified defense that mathematically guarantees that it can withstand all adversarial perturbations within a specific radius and basic defense method without adversarial examples for training. By comparing this method with PM, we show that PM is more effective than other defense methods that do not use adversarial examples.
**RSE:Random Self-Ensemble [44]**
This method is a type of ensemble smoothing that averages predicted results over multiple models or inputs. This method adds a random noise layer to the neural network and ensembles multiple predictions to improve robustness against adversarial attacks. We adopted it for the same reasons as RS.
**AT: Adversarial Training (ideal)**
This method involves the MLP model trained by adversarial training using each attack. AT (ideal) was trained with features extracted by the CLIP model from adversarial examples created using each attack against images in the training set. Note that “ideal” represents each attack to be used for the evaluation, and the obtained results correspond to the upper limit.

When using various adversarial attacks, we evaluated the robustness against adversarial attacks by comparing PM with PM w/o fMRI and PM w/o fMRI with noise. When using clean images, we evaluated the suppression of accuracy degradation by comparing PM with each AT.

**Evaluation via Image and Text Similarity.** We evaluated similarity using two modalities to evaluate how closely the predicted features resembled the original visual features.


**Image similarity**
For image similarity using ${\mathrm{Sim}}_{\mathrm{image}}$ evaluation, we input clean images and adversarial examples generated by various attacks into each comparison model. The cosine similarity between the predicted features ${\hat{f}}_{j}^{o}$ and the original visual features $f_{j}^{o}$ extracted using the CLIP vision encoder was then calculated, and the average of each value for all test datasets *S* was computed. This analysis enables us to evaluate how brain activity influences image features. The equation is expressed as follows:(4)$${\mathrm{Sim}}_{\mathrm{image}}=\frac{1}{S}\sum_{j=1}^{S}\left(\frac{f_{j}^{o^{⊤}}{\hat{f}}_{j}^{o}}{|f_{j}^{o}\left|\right|{\hat{f}}_{j}^{o}|}\right).$$
**Text similarity**
For text similarity using ${\mathrm{Sim}}_{\mathrm{text}}$ evaluation, we computed the cosine similarity between the predicted features ${\hat{f}}_{j}^{o}$ of each comparison model and the text features $f_{j}^{t}$ extracted from the caption corresponding to the original image using the CLIP text encoder and calculated the average of each value for all test datasets *S*. This analysis enables us to evaluate how brain activity influences the relationship between image and text features. The equation is expressed as follows:(5)$${\mathrm{Sim}}_{\mathrm{text}}=\frac{1}{S}\sum_{j=1}^{S}\left(\frac{f_{j}^{t^{⊤}}{\hat{f}}_{j}^{o}}{|f_{j}^{t}\left|\right|{\hat{f}}_{j}^{o}|}\right).$$

**Evaluation via Image-to-Text Retrieval.** To further evaluate the impact of the predicted features on the downstream tasks, we performed quantitative and qualitative evaluations using image-to-text retrieval [45]. In this task, when adversarial examples and clean images are input into each model, the model calculates the cosine similarity between the predicted features and those of the annotated captions in the NSD test set [36] in the CLIP space. Retrieval accuracy was evaluated quantitatively using rank and recall@k metrics.


**Rank**
For each test sample, we determined the ranking position $rank_{j}$ at which the correct caption appeared in the search results and calculated the average of each ranking for all test datasets *S*. Rank ranges from 1 to *S*, and the closer it is to 1, the better the model is at placing the correct caption at a higher position. The equation is expressed as follows:(6)$$\mathrm{Rank}=\frac{1}{S}\sum_{j=1}^{S}\left(rank_{j}\right).$$
**Recall@k**
For each test sample, we calculated the proportion of correct caption $GT_{j}$ appearing within the top-k retrieved results $Result_{j}$ and computed the average of each value for all test datasets *S*. Recall ranges from 0 to 1, and the closer it is to 1, the better the model is at selecting the correct caption within the top-k results. The equation is expressed as follows:(7)$$\mathrm{Recall}@k=\frac{1}{S}\sum_{j=1}^{S}\left(\frac{|GT_{j}\cap Result_{j}|}{|GT_{j}|}\right).$$

In addition, for qualitative evaluation, we visualized the top-1 captions retrieved by each method when processing PGD, APGD, and CLEAN inputs. The motivation for employing this task is to effectively evaluate improvements against attacks that minimize the cosine similarity between the original image and the annotated caption.

#### 4.1.4. Hypothesis Validation

In this experiment, we used the dataset [36], which contains only brain activity data in response to nonattacked images. This is based on the hypothesis that the changes in brain activity caused by adversarial perturbations are minimal, meaning that brain activity would be indistinguishable from that induced by nonattacked images. Figure 4 shows a clean image selected from the test dataset used in this experiment, along with various adversarial examples. All images appear identical; thus, they are confirmed to be indistinguishable. In addition, to support this finding, Table 1 shows the average peak signal-to-noise ratio (PSNR) and structural similarity index measure (SSIM), which are indices for evaluating the image quality of two images, calculated from the original images for the adversarial examples (ϵ=2/255) generated by each attack in this experiment. The smaller the difference between two images, the closer the PSNR to infinity and the SSIM to one. According to a previous study [46], the PSNR and SSIM at which humans are unable to perceive changes in the image are greater than 40 dB and 0.985, respectively. Table 1 shows that all attacks satisfy the condition of being “imperceptible to the human eye’’. As ϵ decreases, the changes in the image reduces, and the PSNR and SSIM increase further. This finding supports the validity of our hypothesis and experimental setup.

#### 4.1.5. Ethical Considerations

The fMRI data used in our experiments are part of the publicly available NSD, which was collected under the approval of an Institutional Review Board (IRB) in the United States. Written informed consent was obtained from all participants, and the data have been anonymized and released in a format suitable for public research use. We exclusively use the anonymized voxel-level fMRI data provided by the NSD and do not access any personally identifiable information. When employing brain signals in the context of adversarial defense, both technical and ethical considerations must be taken into account with great care. In particular, the application of brain activity to improve the robustness of machine learning models is inspired by the inherent noise-resistant nature of human perceptual systems and is positioned within the broader context of socially and medically beneficial research. At the same time, recognizing the potential for misuse or ethical concerns associated with brain data, future studies must adhere strictly to established ethical guidelines and international standards for data protection.

### 4.2. Results and Discussion

To confirm the effectiveness of PM, we present the quantitative and qualitative comparison results.

#### 4.2.1. Evaluation via Image and Text Similarity


**Quantitative Evaluation.** Table 2 presents the image similarity results. PM approaches the upper limit compared with each method. PM reaches approximately 90% of the upper limit for each attack. In addition, when clean images are input, the cosine similarity of each AT is approximately 0.7 at most, whereas PM achieves a very high value of 0.973. Table 3 presents the text similarity results. Although the improvement is smaller than that of the image similarity, it reaches approximately 80% of the upper limit for each attack. In particular, for PGD and APGD with a perturbation of ϵ=2/255, the text similarity is close to the upper limit. When clean images are input, the cosine similarity of each AT is 0.099–0.238, and PM exhibits higher values than CLIP. In terms of both evaluation metrics, a comparison of PM and PM w/o fMRI demonstrates that introducing brain activity in PM is effective.


In addition, by comparing PM and PM w/o fMRI with noise, the effectiveness of introducing brain activity is confirmed. These results demonstrate that PM is robust against adversarial attacks and maintains its accuracy for clean images. By comparing PM and RS or RSE, the results are overwhelmingly superior for all adversarial examples used in the experiment and show that PM is more effective than other defense methods that do not use adversarial examples. Table 4 shows that both AT (PGD) and AT (Gao et al.) are robust only when the attack is used for training, whereas PM exhibits high cosine similarity for a different kind of attack. For example, at ϵ=1/255, AT (PGD) exhibits cosine similarity values of 0.889 against PGD but 0.745 against Gao et al. In addition, AT (Gao et al.) exhibits cosine similarity values of 0.921 against Gao et al. but 0.629 against PGD. However, PM maintains high cosine similarity values of 0.751 against PGD and 0.899 against Gao et al. As shown in Table 5, the text features exhibited a decrease in similarity for samples that were not used during adversarial training. At ϵ=1/255, a similar trend was observed for image similarity. These results demonstrate that PM can obtain robust features even against unknown adversarial attacks.

#### 4.2.2. Evaluation via Image-to-Text Retrieval


**Quantitative Evaluation.** Table 6 presents the quantitative results of image-to-text retrieval. First, for each adversarial attack, all methods improved the text retrieval results compared with CLIP. PM outperformed PM w/o fMRI and PM w/o fMRI with noise for all attacks. In addition, PM approached the upper limit represented by AT (ideal). Next, for clean images, PM outperformed AT, which was trained on each adversarial example. Furthermore, PM closely approached CLIP and performed at a similar level to PM w/o fMRI and PM w/o fMRI with noise; this result demonstrates that PM effectively mitigates accuracy degradation for clean images.**Qualitative Evaluation.** As shown in Figure 5, for the three types of attacks, PM succeeded in retrieving the correct caption for images for which the other comparison methods retrieved completely different results. In addition, PM exhibits successful retrieval in various scenes, not only in specific scenes. In some cases, as shown in Figure 6, PM retrieved captions that were semantically close, although it was not completely successful. Figure 7 presents the top-5 image-to-text retrieval results for PGD. In this figure, PM outputs the correct caption within the top-5 results. This indicates that the PM is able to adequately retain and extract the original semantic features from the brain activity data despite the image being perturbed by the adversarial attack. These results demonstrate that brain activity retains correct caption information and enables caption retrieval at a higher rank.


These evaluation results demonstrate that PM obtains features that are closer to those of the original image. Therefore, combining brain activity with data augmentation allows PM to obtain richer representations. PM achieves robust feature acquisition independent of adversarial attacks without degrading accuracy for clean images.

## 5. Conclusions

In this study, we have proposed an adversarial defense method using brain activity. The contribution of this paper is to demonstrate the effectiveness of brain activity against adversarial attacks against the CLIP model. Experimental results show that PM exhibited superior robustness to adversarial attacks. Furthermore, since PM does not require adversarial examples during training, PM shows excellent robustness not only against the common PGD attack but also against strong PGD-derived attacks and model-independent attacks. It is also found to maintain accuracy even for clean images. These results show that brain activity has robust semantic information and is effective against adversarial attacks. However, there are two challenges. The first is the impracticality of using fMRI data during inference. fMRI is expensive and has high physical costs. Therefore, it is difficult to use fMRI for inference because it is difficult to obtain frequently. The second is the limitation of the fMRI decoder used. MindBridge is personalized to individual brain signals, which makes it difficult to train easily when introducing new subjects and analyze because it is impossible to input each region of the brain. In future work, we are considering multimodal knowledge distillation, in which various brain activity data (fMRI, EEG, etc.) and images are trained in a single model, and weights are transferred so that inference is possible using only one modality (images only, EEG only, etc.). In addition, we are considering overcoming the problem of versatility and enabling analysis in each region of the brain by using a simple and accurate fMRI decoder, which has been actively studied in the reconstruction of visual stimuli. We are also considering creating our own empirical dataset by multisubject experiments that enable zero-shot classification, analyzing the robustness of brain activity in adversarial examples and clean images. 

## Figures and Tables

**Figure 1 sensors-25-02736-f001:**
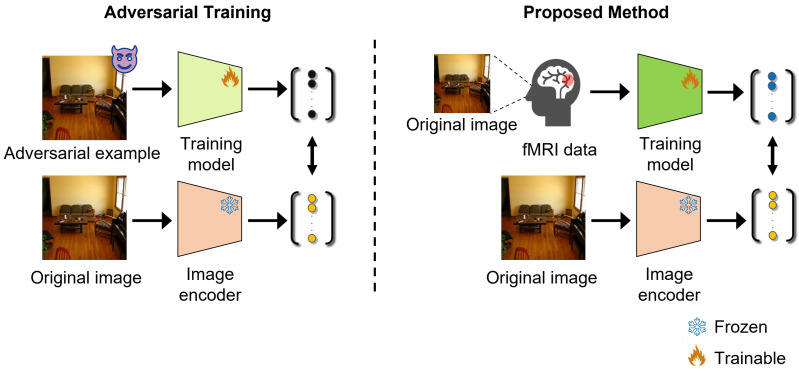
Difference between general adversarial training and the proposed method. Although adversarial training generally uses adversarial examples and original images, the proposed method replaces adversarial examples with brain activity and uses brain activity and original images. The different colors of the training model indicate that the models have the same architecture but different input dimensions.

**Figure 2 sensors-25-02736-f002:**
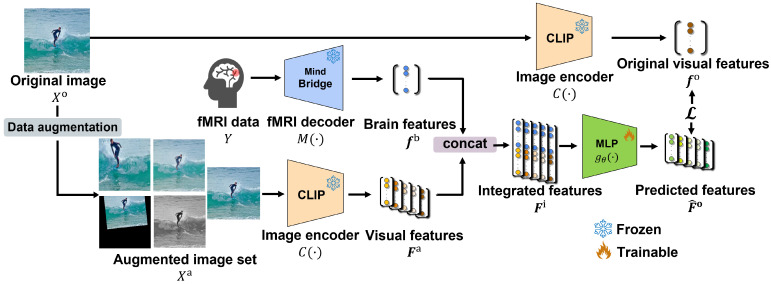
Overview of the proposed method. We train the multilayer perceptron using brain activity features and predict robust features against adversarial attacks in the CLIP space by maximizing the cosine similarity between the predicted and original visual features. The difference in the color of the dots represents the difference in the output features for each input.

**Figure 3 sensors-25-02736-f003:**
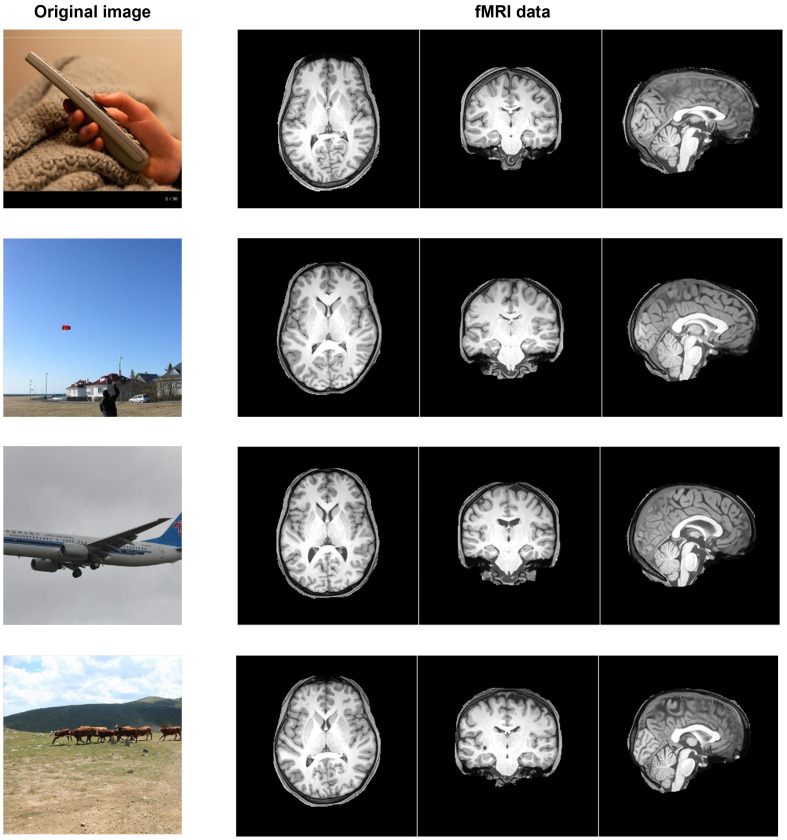
Example of NSD. A paired dataset consisting of original images selected from the MSCOCO dataset and fMRI data capturing human brain activity while viewing those images.

**Figure 4 sensors-25-02736-f004:**
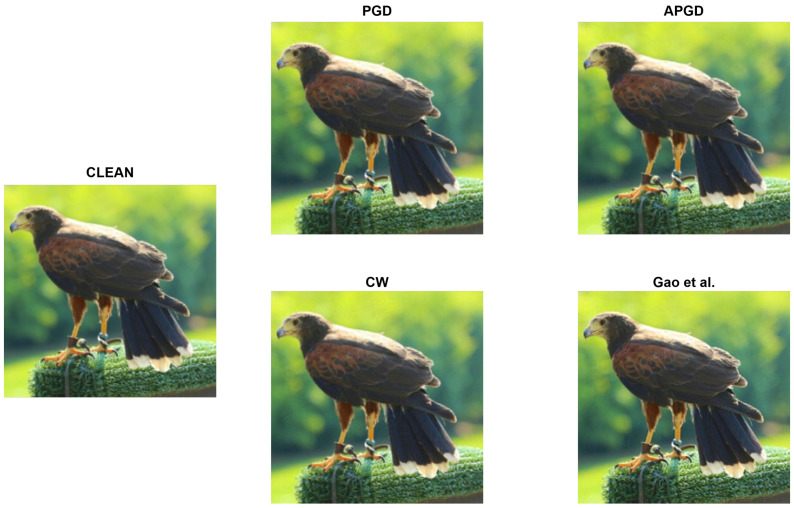
Qualitative image comparison of clean images and various adversarial examples with perturbation of ϵ=2/255. This clean image is included in MSCOCO dataset [37]. The adversarial examples from each attack were created by ourselves [40].

**Figure 5 sensors-25-02736-f005:**
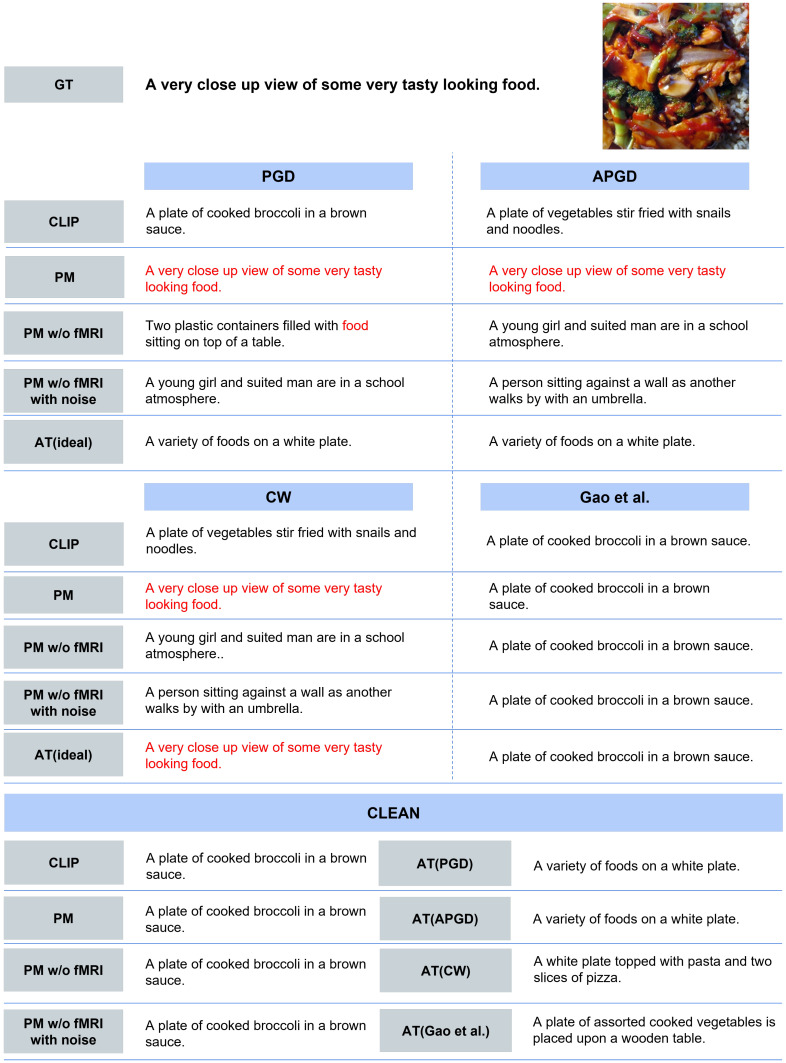
Qualitative evaluation of successful results. The top-1 image-to-text retrieval results when using the adversarial examples created by each attack are presented. Matching words from the ground truth (GT) are shown in red. This image used for this evaluation is included in MSCOCO dataset [37,40].

**Figure 6 sensors-25-02736-f006:**
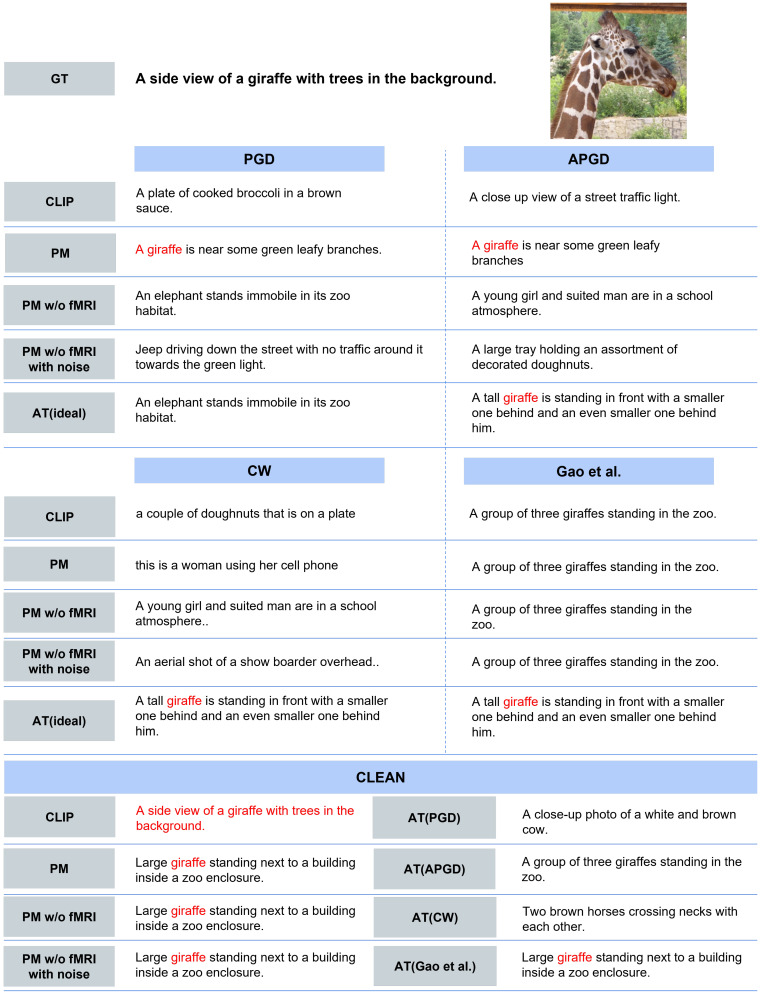
Qualitative evaluation of near-miss results. The top-1 image-to-text retrieval results when using the adversarial examples created by each attack are presented. Matching words from the GT are shown in red. This image used for this evaluation is included in MSCOCO dataset [37,40].

**Figure 7 sensors-25-02736-f007:**
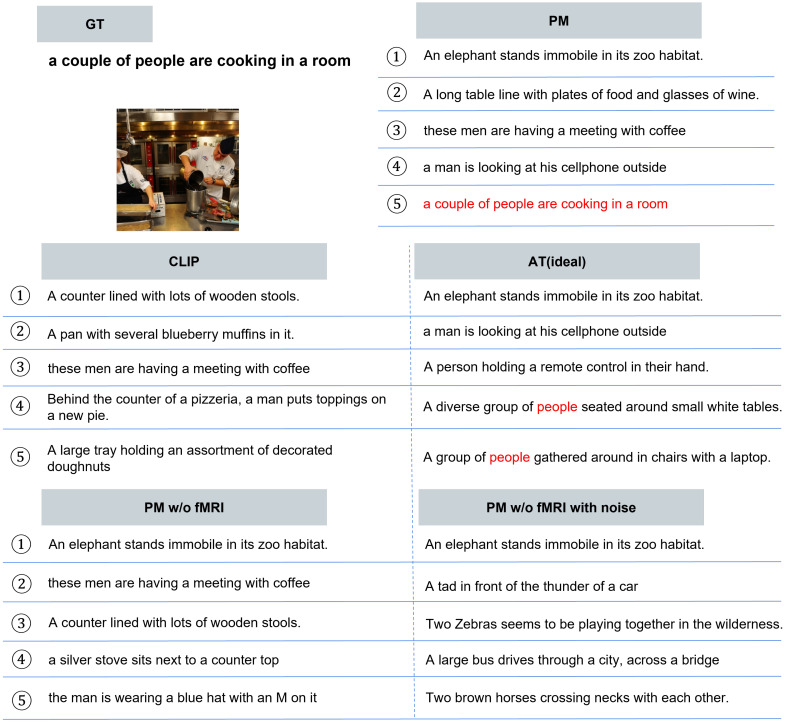
Qualitative evaluation of top-5 results. The top-5 image-to-text retrieval results when using the adversarial example created by PGD are presented. Matching words from the GT are shown in red. This image used for this evaluation is included in MSCOCO dataset [37].

**Table 1 sensors-25-02736-t001:** Averages of PSNR and SSIM values between adversarial examples with ϵ=2/255 created by each attack and the original images $X^{o}$.

	PGD [27]	APGD [28]	CW [29]	Gao et al. [40]	CLEAN
PSNR (dB)	44.6	44.6	43.3	43.5	∞
SSIM	0.988	0.987	0.985	0.989	1.000

**Table 2 sensors-25-02736-t002:** Evaluation of average cosine similarity between predicted and original visual features when several adversarial examples or clean images are input to each method. The results shown in the gray background represent the upper limits against each attack. In the evaluation using clean images, the results of AT (ideal) trained with PGD, APGD, and CW are shown from left to right.

Methods		ϵ=1/255			ϵ=2/255		CLEAN
	PGD	APGD	CW	PGD	APGD	CW	
CLIP	0.799	0.722	0.438	0.477	0.394	0.120	1.00
PM	**0.751**	**0.746**	**0.728**	**0.732**	**0.728**	**0.713**	0.973
PM w/o fMRI	0.557	0.525	0.380	0.428	0.394	0.282	0.991
PM w/o fMRI with noise	0.140	0.128	0.084	0.105	0.093	0.065	0.975
RS [43]	0.383	0.344	0.185	0.236	0.199	0.088	**0.998**
RSE [44]	0.441	0.399	0.223	0.279	0.239	0.112	0.995
AT (ideal)	0.889	0.865	0.800	0.801	0.790	0.794	0.780/0.774/0.681 (ϵ=1/255) 0.724/0.703/0.581 (ϵ=2/255)

**Table 3 sensors-25-02736-t003:** Evaluation of average cosine similarity between predicted and text features.

Methods		ϵ=1/255			ϵ=2/255		CLEAN
	PGD	APGD	CW	PGD	APGD	CW	
CLIP	0.166	0.128	−0.095	0.0112	−0.031	−0.326	0.257
PM	**0.167**	**0.165**	**0.148**	**0.156**	**0.153**	**0.128**	0.258
PM w/o fMRI	0.150	0.135	0.048	0.089	0.071	−0.039	**0.264**
PM w/o fMRI with noise	0.060	0.056	0.031	0.041	0.034	0.005	0.261
RS [43]	0.092	0.074	−0.018	0.027	0.007	−0.104	0.258
RSE [44]	0.111	0.091	−0.011	0.039	0.016	−0.105	0.259
AT (ideal)	0.221	0.207	0.171	0.161	0.166	0.207	0.238/0.227/0.184 (ϵ=1/255) 0.135/0.176/0.099 (ϵ=2/255)

**Table 4 sensors-25-02736-t004:** Evaluation of average cosine similarity between predicted and original visual features. We investigated the attack dependencies of adversarial defense methods against new attacks. The upper limit for each attack is shown in the gray background.

Methods	ϵ=1/255		ϵ=2/255	
	PGD	Gao et al. [40]	PGD	Gao et al. [40]
PM	**0.751**	**0.899**	**0.732**	**0.769**
PM w/o fMRI	0.557	0.898	0.428	0.731
PM w/o fMRI with noise	0.140	0.890	0.105	0.727
RS [43]	0.383	0.864	0.236	0.646
RSE [44]	0.441	0.870	0.279	0.660
AT (PGD)	0.889	0.745	0.801	0.685
AT (Gao et al. [40])	0.629	0.921	0.636	0.849

**Table 5 sensors-25-02736-t005:** Evaluation of average cosine similarity between predicted and text features. We investigated the attack dependencies of adversarial defense methods against new attacks.

Methods	ϵ=1/255		ϵ=2/255	
	PGD	Gao et al. [40]	PGD	Gao et al. [40]
PM	**0.167**	**0.210**	**0.156**	**0.143**
PM w/o fMRI	0.150	0.204	0.0894	0.121
PM w/o fMRI with noise	0.0600	0.208	0.0410	0.127
RS [43]	0.092	0.186	0.027	0.090
RSE [44]	0.111	0.190	0.039	0.095
AT (PGD)	0.221	0.207	0.161	0.157
AT (Gao et al. [40])	0.152	0.227	0.130	0.192

**Table 6 sensors-25-02736-t006:** Quantitative results of image-to-text retrieval. AT (ideal) is the method of adversarial training using each attack used in the evaluation, and the results shown in the gray background represent the upper limits.

	Methods	Rank	Recall@1	Recall@5	Recall@10
PGD	CLIP	494.31	0.011	0.033	0.056
	PM	**197.17**	**0.019**	**0.068**	**0.103**
	PM w/o fMRI	461.82	0.005	0.014	0.021
	PM w/o fMRI with noise	481.39	0.001	0.004	0.009
	AT (ideal)	144.94	0.011	0.059	0.111
APGD	CLIP	651.23	0.003	0.011	0.020
	PM	**214.06**	**0.015**	**0.056**	**0.098**
	PM w/o fMRI	541.91	0.001	0.010	0.015
	PM w/o fMRI with noise	513.71	0.001	0.006	0.008
	AT (ideal)	154.20	0.012	0.058	0.126
CW	CLIP	949.20	0.000	0.000	0.002
	PM	**346.70**	**0.006**	**0.026**	**0.048**
	PM w/o fMRI	866.49	0.000	0.000	0.002
	PM w/o fMRI with noise	707.11	0.000	0.001	0.002
	AT (ideal)	42.50	0.090	0.325	0.486
Gao et al. [40]	CLIP	210.78	0.012	0.063	0.116
	PM	**117.71**	**0.040**	**0.131**	**0.204**
	PM w/o fMRI	167.65	0.022	0.098	0.159
	PM w/o fMRI with noise	158.36	0.037	0.106	0.169
	AT (ideal)	67.12	0.048	0.176	0.296
CLEAN	CLIP	4.96	0.485	0.771	0.872
	PM	5.23	0.461	0.756	**0.860**
	PM w/o fMRI	**5.10**	**0.474**	**0.763**	0.859
	PM w/o fMRI with noise	5.21	0.467	0.762	**0.860**
	AT (PGD)	254.13	0.005	0.028	0.061
	AT (APGD)	42.66	0.061	0.225	0.366
	AT (CW)	352.05	0.001	0.020	0.038
	AT (Gao et al. [40])	7.44	0.311	0.641	0.777

## Data Availability

Data are contained within the article.

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
