# Peer review of "Enhancing Adversarial Defense via Brain Activity Integration Without Adversarial Examples"

_sensors, 2025, doi:10.3390/s25092736_

Round 1

Reviewer 1 Report

Comments and Suggestions for Authors

This paper proposes a novel adversarial defense method for the CLIP model that integrates human brain activity data (fMRI) with data-augmented images to enhance robustness against adversarial attacks. The proposed method aims to leverage the inherent robustness of human perception by training a model to maximize the similarity between combined features and the original image's features in CLIP space. Experimental results demonstrate that this approach significantly enhances robustness against various adversarial attacks without degrading accuracy on clean images, overcoming the typical trade-off seen in adversarial training.

Strength

+ The integration of human brain activity data for adversarial defense in VLMs like CLIP is novel and interesting. It explores a fundamentally different direction compared to traditional defense mechanisms.

+ The method directly targets major drawbacks of standard adversarial training: the need for prior knowledge of attack types and the often-observed degradation of performance on clean data.

+ The experiments show substantial improvements in robustness metrics (image/text similarity, retrieval rank/recall) under various attacks compared to baseline CLIP and ablation studies (PM w/o fMRI). Crucially, performance on clean data remains high, close to the original CLIP, and significantly better than AT(ideal).

Main concerns

- One of my most significant concerns is the reliance on fMRI data. Acquiring fMRI data is extremely expensive, requires specialized hardware and expertise, is time-consuming, and typically yields data from a very small number of subjects and stimuli compared to standard image datasets. This severely limits the practicality, scalability, and generalizability of the proposed method for real-world applications or training on large-scale models/datasets.

- The experiments rely on fMRI data from a single subject in the NSD dataset. Brain activity patterns can vary significantly across individuals. It is unclear how well the learned model or the approach itself would generalize if trained or tested with data from different subjects. The findings might be specific to the brain patterns of this one individual.

- The effectiveness of the proposed method appears heavily dependent on the quality and availability of the fMRI decoder (MindBridge [35]) used to translate brain activity into the CLIP feature space. The sensitivity to the choice and performance of this decoder is not explored. Reproducibility might be an issue if this decoder is not readily accessible or easily trainable.

- While feature similarity and retrieval are relevant, evaluating robustness directly on downstream task performance, such as zero-shot classification accuracy under attack (e.g., on ImageNet), would provide stronger evidence of practical defense capability.

Questions for authors

1. Could you elaborate on the challenges and limitations imposed by using single-subject fMRI data? How might the results change if multi-subject data were available, potentially using subject-independent fMRI decoding techniques?

2. How critical is the choice of the MindBridge [35] fMRI decoder? Have you considered or experimented with alternative methods for mapping fMRI signals to the CLIP space, perhaps simpler ones, and how does this impact performance?

3. Would it be possible to evaluate the proposed method on a standard zero-shot classification benchmark (like ImageNet) under both clean and adversarial conditions (using the generated robust features) to further assess its practical impact on core CLIP capabilities?

Author Response

We appreciate your review of the paper.
Please see the attachment. 

Reviewer 2 Report

Comments and Suggestions for Authors

The authors propose a adversarial defense method for vision-language foundation models that integrates human brain activity data (fMRI) with image augmentation to learn robust feature representations without the need for adversarial examples. The method leverages the assumption that adversarial examples, which are visually indistinguishable from clean images to humans, do not significantly alter neural representations in the human brain. The authors demonstrate that this integration allows the model to learn visual features that are both robust to various adversarial attacks and accurate on clean images. Based on experiments conducted using the Natural Scenes Dataset (NSD), the authors claim that their method is effective in preserving image-text similarity and improving image-to-text retrieval performance across multiple adversarial attack scenarios. The idea is interesting. There are the areas of improvement.

  1. Please discuss implications of inter-subject variability and consider fine-tuning or calibrating across multiple users in future work.
  2. Explore surrogate modeling (e.g., simulated brain activity) or EEG data for practical implementations.
  3. The paper lacks comparisons with some newer universal defense mechanisms like ensemble smoothing or certified defenses.
    Please offer a discussion of these alternatives.
  4. Perform region-specific fMRI analysis to validate the cognitive relevance of robustness.
  5. Include a short discussion about ethical considerations, especially in terms of data privacy and consent.
  6. Clarify what kinds of attacks were tested and limit claims to tested domains.
  7. The assumption that brain responses are invariant to adversarial perturbations must be empirically validated, ideally by measuring brain activity in response to adversarial and clean stimuli and comparing them statistically.
  8. Explore lightweight alternatives or precomputed embeddings to reduce implementation complexity.
  9. Include a formal analysis or theoretical model that explains why brain-derived features offer adversarial robustness
  10. Expand experimental comparisons to include contemporary adversarial defense techniques such as self-supervised adversarial training, certified defenses, and test-time adaptation or adversarial purification.
  11. The conclusions section is very short and does not reflect the results, expand it.
  12. Figure 7 is very short and it is strange for scientific paper.

Author Response

(The authors gave the same response as above.)

Round 2

Reviewer 1 Report

Comments and Suggestions for Authors

The authors have substantially revised most of my prior concerns and comments. This manuscript is recommended for acceptance. Thank you.

Reviewer 2 Report

Comments and Suggestions for Authors

The paper is improved.